# Bioinformatic Multi-Strategy Profiling of Congenital Heart Defects for Molecular Mechanism Recognition

**DOI:** 10.3390/ijms252212052

**Published:** 2024-11-09

**Authors:** Fabyanne Guimarães de Oliveira, João Vitor Pacheco Foletto, Yasmin Chaves Scimczak Medeiros, Lavínia Schuler-Faccini, Thayne Woycinck Kowalski

**Affiliations:** 1Laboratory of Medical Genetics and Evolution, Graduate Program in Genetics and Molecular Biology, Genetics Department, Universidade Federal do Rio Grande do Sul (UFRGS), Porto Alegre 91501-970, Brazil; fabyanneoliveira@hcpa.edu.br (F.G.d.O.); jfoletto@hcpa.edu.br (J.V.P.F.); lavinia.faccini@ufrgs.br (L.S.-F.); 2Teratogen Information System (SIAT), Medical Genetics Service, Hospital de Clínicas de Porto Alegre (HCPA), Porto Alegre 90035-903, Brazil; ycmedeiros@hcpa.edu.br; 3Laboratory of Genomic Medicine, Center of Experimental Research, Hospital de Clínicas de Porto Alegre (HCPA), Porto Alegre 90035-903, Brazil; 4National Institute on Population Medical Genetics (INAGEMP), Porto Alegre 90035-903, Brazil; 5Graduate Program in Children and Adolescent Health, Medicine Faculty, Universidade Federal do Rio Grande do Sul (UFRGS), Porto Alegre 90035-903, Brazil; 6Bioinformatics Core, Hospital de Clínicas de Porto Alegre (HCPA), Porto Alegre 90035-903, Brazil; 7Graduate Program in Medicine: Medical Sciences, Medicine Faculty, Universidade Federal do Rio Grande do Sul (UFRGS), Porto Alegre 90035-003, Brazil

**Keywords:** tetralogy of Fallot, systems biology, transcriptome, RNA-seq, microarray, ontologies, cardiogenesis

## Abstract

Congenital heart defects (CHDs) rank among the most common birth defects, presenting diverse phenotypes. Genetic and environmental factors are critical in molding the process of cardiogenesis. However, these factors’ interactions are not fully comprehended. Hence, this study aimed to identify and characterize differentially expressed genes involved in CHD development through bioinformatics pipelines. We analyzed experimental datasets available in genomic databases, using transcriptome, gene enrichment, and systems biology strategies. Network analysis based on genetic and phenotypic ontologies revealed that *EP300*, *CALM3*, and *EGFR* genes facilitate rapid information flow, while *NOTCH1*, *TNNI3*, and *SMAD4* genes are significant mediators within the network. Differential gene expression (DGE) analysis identified 2513 genes across three study types, (1) Tetralogy of Fallot (ToF); (2) Hypoplastic Left Heart Syndrome (HLHS); and (3) Trisomy 21/CHD, with *LYVE1*, *PLA2G2A*, and *SDR42E1* genes found in three of the six studies. Interaction networks between genes from ontology searches and the DGE analysis were evaluated, revealing interactions in ToF and HLHS groups, but none in Trisomy 21/CHD. Through enrichment analysis, we identified immune response and energy generation as some of the relevant ontologies. This integrative approach revealed genes not previously associated with CHD, along with their interactions and underlying biological processes.

## 1. Introduction

Congenital heart defects (CHDs) are anatomical malformations of the heart and/or major vessels that occur during the embryonic period [1]. Twenty-eight percent of all severe congenital anomalies consist of heart defects, representing a significant global health problem [2]. It occurs in 0.8–1 of every 100 live births, with 25% of diagnosed CHDs needing surgery or intervention, leading to a risk of death within the first month of birth [3,4]. CHDs can be classified into eight categories: conotruncal defects (Tetralogy of Fallot, double outlet right ventricle, and D-transposition of the great arteries), atrioventricular septal defects, left ventricular outflow tract obstructions (hypoplastic left heart syndrome, aortic stenosis, and coarctation of the aorta), septal defects (ventricular and atrial), right ventricular outflow tract obstructions (tricuspid and pulmonary atresia), heterotaxic malformations, complete defects (L-transposition of the great arteries and other defects), and anomalous pulmonary venous return (total or partial) [5].

The etiology of these defects is multifactorial and includes environmental factors, which may contribute to 10% of CHDs, as well as genetic causes, which may be associated with syndromes or occur as isolated heart defects [6]. Maternal diabetes and obesity, alcohol exposure, congenital infections (rubella, hepatitis B), and certain medications (lithium, isotretinoin) are considered environmental factors that increase the risk of congenital heart defects [7,8,9,10,11]. Approximately 70% of CHDs occur in isolation, having a multifactorial etiology; this includes the most severe cardiac defects. Still, they can also be associated with other congenital defects or as part of known genetic syndromes, which usually have a known etiology, such as chromosomal, monogenic, and/or teratogenic causes [12]. Genetic contributions to the development of CHDs vary widely, i.e., CHDs are diagnosed in 35% to 50% of newborns with Trisomy 21, in 60% to 80% with Trisomy 18 and 13, and in 33% with Monosomy X [12,13]. Copy number variations (CNVs) also influence CHDs and can either occur de novo or be inherited. Many CNVs are related to clinically recognized syndromes, such as 22q11.2 deletion syndrome, 7q11.23 deletion (Williams–Beuren syndrome), and 5p15.2 deletion (Cri-Du-Chat syndrome) [13,14,15]. De novo mutations (DNMs) are found in genes highly expressed during heart development, with DNMs being linked to approximately 10% of congenital heart defects [16,17].

The advances in sequencing technologies have helped increase the understanding of the human genome and, consequently, contributed to the discovery of new candidate genes associated with CHDs, facilitating the identification of their genetic variants [18]. It is estimated that the use of molecular biology techniques, bioinformatics tools, and the availability of population study databases have enabled the identification of pathogenic variants in definitive and candidate genes for congenital heart defects in 45% of affected patients [16,19]. The functional characterization of a gene includes identifying its interactions, both at genomic and protein levels. Systems biology is an integrated approach that provides a comprehensive view of genes, proteins, and their interactions through models that assess disturbances, proper phenotype evaluation, and computational methods for probabilistic and mathematical modeling. This approach can assist in understanding gene interactions and the etiology of congenital heart defects [20,21]. Genetic and phenotypic ontology databases have been widely used to explore these molecular mechanisms for characterizing phenotypes and gene functions [22,23].

Based on this approach, the present study aimed to identify and characterize genes that have potential roles in heart development and are consequently involved in the molecular mechanisms related to congenital heart defects. Considering that these defects are prevalent in newborns with diverse phenotypes, in silico analyses were conducted using systems biology tools to investigate the relationship between genes involved in cardiac development and CHDs through genetic and phenotypic ontologies.

## 2. Results

A scheme of the strategy developed in the present study is described in Figure 1.

### 2.1. Gene and Phenotype Ontology Analysis and Network Statistics

The GO selection yielded 651 genes corresponding to 430 previously selected ontologies (Appendix A), while the 19 CHD phenotypes provided 1111 genes through the Human Phenotype Ontology (HPO) (Appendix A). A Venn diagram revealed that 177 genes are shared between both repositories (Appendix A). Some of these genes are known to play critical roles at various stages of cardiac development, including *NKX2-5*, *T-box* family genes (*TBX1*, *TBX2*, and *TBX5*), *GATA* family genes (*GATA4*, *GATA5*, and *GATA6*), and *MYH6*. Among the genes identified in the HPO, 934 (84%) have not been previously associated with CHDs; however, they are involved in other biological processes such as peroxisome maintenance and organization, cilium assembly and organization, and microtubule-based transport and movement. These processes are critical for the formation of a healthy heart, and their disruption can directly or indirectly influence the development of CHDs. Using the *STRING* tool, we analyzed the GO and HPO networks individually and in combination, applying the resulting network to Cytoscape v.3.10.0 for topological analysis (Appendix A). To better illustrate gene interactions, a network featuring the genes shared between the HPO and GO is also shown (Figure 2).

Following the same approach, we combined the common HPO and GO genes into a network, identifying the main nodes involved in information flow. Sixty-one genes (nodes) exhibited at least one interaction (edge), with a clustering coefficient of 0.151 and an average number of neighbors of 3.115. This metric means that each node is connected, on average, with at least three other nodes. Based on this analysis, *EP300*, *CALM3*, and *EGFR* had the highest betweenness and closeness centrality levels, meaning that they have high and fast information flow. Additionally, the genes *NOTCH1*, *TNNI3*, and *SMAD4* emerged as significant mediators of information flow to other genes within the network and have been shown to have a known relationship with CHDs. However, to better understand the gene interactions involved in CHDs, we chose to search for additional genes that could contribute to the various CHD phenotypes.

### 2.2. Differential Gene Expression Analysis

A total of 122 gene expression studies were obtained through a search in the Gene Expression Omnibus (GEO) repository, selected by two authors based on the inclusion criteria. Among these, 35 studies (28.6%) were excluded for not comprising data on CHDs. Out of the 87 studies (71.4%) related to CHDs, 31 studies (35.6%) did not use microarray or bulk RNA-Seq gene expression methodologies, 17 studies (19.5%) had less than four samples for cases and/or controls, 13 studies (15.0%) were knockout, 6 studies (6.9%) lacked a control group, and another 6 studies (6.9%) did not include cardiac samples. Thus, these studies were excluded from the analysis. Fourteen studies (16.1%) met the selection criteria; however, only seven were included in the final analysis due to processing limitations: GSE196443 (Trisomy 21/CHD), GSE217557 (Trisomy 21/CHD), GSE141955 (Tetralogy of Fallot (ToF)), GSE132401 (Tetralogy of Fallot (ToF) and Single Ventricle Disease (SVD)), GSE36761 (Tetralogy of Fallot—ToF), GSE23959 (Hypoplastic Left Heart Syndrome (HLHS)), and GSE209677 (cardiac cell differentiation). Details about these selected studies are available in Appendix A.

The selected studies were analyzed individually, resulting in 2513 differentially expressed genes across the six studies, as shown in Appendix A. The dataset of SVD diagnosis samples (GSE141955) did not present significantly differentially expressed genes and was excluded from further analysis. The Lymphatic Vessel Endothelial Hyaluronan Receptor 1 (*LYVE1*), which encodes a membrane glycoprotein involved in hyaluronan transport during various stages of cell growth, was found to be differentially expressed in three studies: GSE23959 (HLHS), where it was upregulated, and in ToF, on GSE141955 and GSE36761, where it was downregulated. Similarly, a member of the phospholipase A2 family, *PLA2G2A*, was also differentially expressed, with downregulation observed across these studies. Additionally, a member of the short-chain dehydrogenase/reductase enzyme family, *SDR42E1*, was found to be upregulated in three of the analyzed studies: GSE36761 (ToF), GSE132401 (ToF), and GSE217557 (Trisomy 21/CHD). Two studies highlighted genes involved in cell recognition: the Mannose Receptor C-Type 1 (*MRC1*), which encodes membrane receptors involved in the endocytosis process, was downregulated in both GSE141955 (ToF) and GSE36761 (ToF); while the Potassium Two Pore Domain Channel Subfamily K Member 3 (*KCNK3*), a member of the potassium channel protein superfamily, was downregulated in GSE36761 (ToF) and GSE132401 (ToF). The ADAM Metallopeptidase with Thrombospondin Type 1 Motif 9 (*ADAMTS9*) gene, which plays a role in proteoglycan cleavage and organ morphology regulation during development, was significantly downregulated in two studies: GSE23959 (HLHS) and GSE36761 (ToF).

To better comprehend the impact of the genes identified in the process of heart development, differential gene expression was analyzed in a healthy cardiac cell differentiation dataset (GSE209677) and compared with previously identified genes in ToF, HLHS, and Trisomy 21 studies to determine if these genes’ expressions were altered through the differentiation process. Using a Venn diagram, 578 common genes were identified, including *PLA2G2A* and *ADAMTS9*, which were significant in studies involving ToF and HLHS, and *KCNK3*, which was significant only in ToF studies.

Additionally, we compared the differentially expressed genes shared across all datasets with those identified through ontology searches in the GO and HPO. We grouped the studies into three categories based on CHD diagnosis: (1) studies on ToF; (2) the study on HLHS; and (3) studies on Trisomy 21 with CHDs (Table 1). Using the systems biology approach, we observed that in the HLHS (GSE23959) and ToF (GSE36761) datasets, there were interactions between the nodes representing the differentially expressed genes from each study and the genes found in the GO and HPO. In contrast, for Trisomy 21 with the CHD dataset, no interaction was detected between the networks (Figure 3). Through the combined analyses, we identified genes that may contribute to the development of CHDs and characterized the interactions and biological processes in which they are involved.

### 2.3. Ontology Enrichment Analysis

The genes identified in the DGE analyses that were not common to the GO and HPO searches were further analyzed to verify their associated ontologies and signaling pathways. As mentioned, the datasets were divided into three groups according to the type of CHD. The ontologies identified for the ToF studies (GSE141955, GSE132401, and GSE36761) were related to various immunologic processes, including leukocyte-mediated immunity, with 109 genes involved in this process, such as *C1RL*, *IGLC7*, and *IGHV3-33*; and the adaptive immune response based on somatic recombination of immune receptors derived from immunoglobulin superfamily domains, with 98 associated genes, including *CCL19*, *C3*, and *IGHG2*. The regulation of cell–cell adhesion was also enriched, with 76 genes involved, such as *LILRB2*, *ITGB2*, and *IL1RN* (Appendix A).

For the HLHS study (GSE23959), the ontologies were related to biological processes, such as the generation of precursor metabolites and energy, with 20 associated genes including *ATP5MG*, *NDUFB9*, and *NDUFA8*; energy derivation by oxidation of organic compounds, with 16 genes involved such as *COQ10A*, *UQCRB*, and *NDUFAF1*; and aerobic respiration, with ten genes involved, including *PANK2*, *NDUFA9*, and *SUCLA2* (Appendix A). Ontologies such as response and defense against viruses, antimicrobial humoral immune responses mediated by antimicrobial peptides, and the regulation of myoblast fusion were associated with 20 genes in the Trisomy 21 with CHDs datasets (GSE196443 and GSE217557), including GNLY, *IFI44L*, and *CXCL9* (Appendix A). The genes associated with the observed ontologies were also compared with the dataset related to the differentiation of healthy cardiac cells (GSE209677). Out of the 587 related genes, *IL1RN* (ToF) and *IFI44L* (Trisomy 21/CH) were linked to cardiac cell differentiation. Therefore, it is possible to observe that the ontologies related to differentially expressed genes not yet described in the GO and HPO, are responsible for biological processes that directly influence cardiac development, and when affected, may contribute to alterations in this stage of heart formation.

## 3. Discussion

Heart development is mediated by several biological mechanisms, involving many genes and environmental factors. Through the search for genetic (651 genes) and phenotypic (1111 genes) ontologies, we identified 177 common genes already described as contributing to CHDs, including *EP300*, *CALM3*, *EGFR*, *NOTCH1*, *TNNI3*, and *SMAD4*. However, CHDs present high clinical variability even in recognized syndromes, as they can be associated with congenital or isolated defects. Their complex etiology directly influences the phenotype presented. Hence, we expanded the search for genes that may alter the disease phenotype using DGE analysis. This strategy resulted in 2513 differentially expressed genes obtained from six datasets, divided into three categories based on CHD diagnosis: ToF, HLHS as an isolated CHD, and a dataset of Trisomy 21 associated with CHDs. The genes *LYVE1*, *PLA2G2A*, and *SDR42E1* were deregulated in three of the six analyzed studies, followed by *MRC1*, *KCNK3*, and *ADAMTS9*, which were differentially expressed in two of the six analyzed studies. All differentially expressed genes, categorized by CHD diagnosis, were compared with a dataset of healthy cardiac cells, where we identified 578 common genes between the two datasets, including the aforementioned *PLA2G2A*, *ADAMTS9*, and *KCNK3* genes. The genes identified in the present study are listed in Table 2. Additionally, with a systems biology approach, the genetic and phenotypic ontologies identified were compared with the genes differentially expressed. Datasets on isolated CHDs (ToF and HLHS) showed greater interaction with the GO and HPO data, whereas CHDs associated with Trisomy 21 showed no interaction between genes from the repositories. Having identified that genes common to the GO and HPO databases are involved in the development of CHDs, we sought to determine how the genes not found in these GO and HPO repositories contribute to CHDs. Therefore, using gene ontology overrepresentation analysis, we found that most of these genes are involved in immunological processes, energy generation, secondary metabolite production, and cellular communication, which are fundamental for broad cellular maintenance. The identification of these genes that do not directly act on cardiac development might be a consequence of the bioinformatics approach used, which identifies a wide range of genes, including those with many cellular functions. Therefore, to verify the specific contribution of these genes to the development of CHDs, experimental validation is necessary, which constitutes a limitation of the study.

To propel the understanding of congenital heart disease (CHD), it is essential to explore both embryology and associated genetic factors. Such exploratory approaches can be performed, using available resources such as the GO and HPO. In this study, we used these resources to identify previously described gene and phenotypic ontologies for CHD. The *NKX2-5* gene stands out as it is involved in multiple stages of heart formation, acting as a key marker in the differentiation of cardiac precursor cells, including the development of the conduction system, valves, and cardiac septal [24]. This gene works with other highly conserved transcription factors (such as *TBX20*, *GATA4*, and *MYH6*) and has a central role in organizing the process of cardiogenesis [25]. Variations in genes known to regulate cardiac development have been extensively studied in both isolated and syndromic CHD, and have been linked to numerous phenotypes, including ToF, atrial septal defect, and ventricular septal defect [26].

Using systems biology, we integrated the genes identified in the GO and HPO to provide new insights into CHD, revealing genes that might play key roles in the flow of biological information. The *EP300*, critical for cell regulation and differentiation through chromatin remodeling, already has pathogenic variants associated with ToF [26]. *EGFR* encodes tyrosine kinase receptors and is essential for cardiac cell development, specialization, and differentiation, particularly in regulating human aortic valve embryogenesis [27,28]. *CALM3* encodes a highly conserved protein expressed in the heart, regulating various ion channels in cardiac cells, and is involved in several biological processes, such as muscle contraction, inflammation, metabolism, and immune responses [29]. While EP300, EGFR, and CALM3, along with other identified genes, have been associated with CHD, it is crucial to continue identifying genetic variants. Pathogenic variants in specific genes may interfere with embryonic viability and development, potentially leading to embryonic lethality, as seen with variants in the EGFR gene [28]. We also identified and characterized genes involved in three different CHD phenotypes through DGE analysis of six datasets available in the GEO repository. This analysis revealed 2513 differentially expressed genes, including *LYVE1* and *PLA2G2A*, which were found in two of the ToF studies and in the HLHS study. Notably, these genes were not previously mentioned in the literature in relation to CHD. *LYVE1* was described as a marker in abnormal lymphatic system development studies, particularly in congenital diaphragmatic hernia [30], and increased nuchal translucency, as seen in Noonan syndrome cases [31]. *PLA2G2A* has been identified in studies focused on tumors [32] and diabetes [33]. The *SDR42E1* gene expressed in two ToF studies and in one Trisomy 21/CHD study plays a crucial role in vitamin D biosynthesis [34], with variations in this gene linked to fragile cornea syndrome [35]. The *MRC1* gene, involved in biological processes such as chemotaxis and leukocyte migration, is associated with cleft palate, another congenital defect [36], and was expressed in two ToF studies. The *KCNK3* and *ADAMTS9* genes have been implicated in cardiac function, with *KCNK3* identified as a predisposing factor for pulmonary arterial hypertension (PAH) [37], primarily affecting atrial function and playing roles in rhythm regulation and cardiac conduction [38]. *ADAMTS9* is essential for proper cardiovascular development and adult homeostasis, with expression in derivatives of the secondary heart field, vascular smooth muscle cells in the arterial wall, mesenchymal cells of the valves, and non-myocardial cells of the ventricles. It has a described association with CHDs such as bicuspid aortic valve disease [39,40], although it was not found to be related to HLHS and ToF.

With all these genes identified and based on the indication that they play a role in the development of CHD, we aimed to compare these results with datasets of healthy cardiac cells. This comparison revealed 578 genes shared among all analyzed datasets and corroborated the roles of *KCNK3* and *ADAMTS9* in biological processes relevant to cardiac embryogenesis [37,39,40]. Additionally, we found that while genes such as *PLA2G2A* are related to heart formation, their specific roles in the development of CHD remain unclear. Considering the broad range of genes not yet directly linked to CHD, we integrated the datasets from the three CHD groups with the data identified in the GO and HPO. Through a protein interaction network analysis, we observed that in the ToF and HLHS groups, there were interactions between nodes related to the disease and those in the GO and HPO; however, this pattern was not evident in the Trisomy 21/CHD group.

ToF, the most common cyanotic CHD, is characterized by a ventricular septal defect, right ventricular outflow tract obstruction, an overlapping of the ventricular septum by the aortic root, and right ventricular hypertrophy [41]. Multiple transcription factors and signaling molecules are related to the disease as reported in the literature, including *GATA4*, *NKX2-5*, *JAG1*, *FOXC2*, *TBX5*, and *TBX1*, which is consistent with the interactions observed with the genes listed in the repositories [42]. Ventricular function is affected in ToF, which may influence gene expression during development. We observed this difference in our results, particularly in the analysis of GSE132401, which studied induced pluripotent stem cells and presented a small number of differentially expressed genes compared to GSE36761, which analyzed ventricular samples. HLHS, on the other hand, is a severe cyanotic CHD resulting from the underdevelopment of the left ventricle, mitral valve, aortic valve, and ascending aorta. Numerous genetic variants and molecular pathways in HLHS have been discovered in recent decades, which was reflected in the interactions identified in the network [43].

The same interaction pattern was not observed in the studies of Trisomy 21 and CHD. Despite CHD being one of the main causes of morbidity and mortality, and the presence of Trisomy 21 being associated with a 50-fold greater likelihood of developing CHD compared to the general population [44], we failed to identify interactions in the repositories used for this study. This indicates a need for new approaches to elucidate the various phenotypes of CHD, especially considering the different etiologies.

Therefore, analyzing the differentially expressed genes from datasets that did not overlap with the GO and HPO was essential for understanding their involvement in congenital heart disease (CHD) through gene ontologies. We observed various genes associated with the immune response across all analyzed datasets. This may be related to multiple biological processes occurring during the embryonic period, such as placental formation, which develops alongside the heart and acts as a barrier that regulates nutrient and oxygen transfer while preventing the passage of pathogens and cells that could impair development [45]. Placental dysfunction has been linked to poor cardiac development, as previously described in the literature [46,47]. Also, we identified genes enriched in ontologies related to the generation of precursor metabolites and energy, which have been associated with the development and postnatal functions of the right ventricle in newborns with CHD. Changes in the ventricular phenotype, such as volume overload in HLHS, could lead to serious consequences that are not yet fully understood [48].

## 4. Materials and Methods

### 4.1. Selection of Ontologies for CHD

The complete list of gene ontologies (GOs) and phenotype ontologies was obtained from the AmiGO and Human Phenotype Ontology (HPO) databases, respectively. The GO search was performed using keywords (“cardiac”, “heart”, “cardio”, “myocardial”, “myocardium”, “atrium”, “atrial”, “ventricular”, “ventricle”, “septum”, “septal”, “valve”) with the GO.db package (R v.4.3.1), which identified 727 ontologies, of which 430 were selected for the next step. This selection was conducted by two independent authors who reviewed the ontologies and retained those considered relevant.

With respect to the HPO, the selection was based on the CHDs diagnosed at the Hospital de Clínicas de Porto Alegre, according to a project on active vigilance of congenital anomalies, based on Cardoso et al., 2021 [49]. A total of 2448 genes were found for 19 phenotypes. After removing duplicate genes, 1111 genes were selected for further analysis. To better visualize the selected genes from both ontologies, Venn diagrams were generated throughout the study (https://bioinformatics.psb.ugent.be/webtools/Venn/, accessed on 15 September 2024) [50].

### 4.2. Systems Biology Analysis

The genes selected from the ontology analyses were input into the STRING v.12 tool [51], where networks of protein–protein interactions (PPIs) for Homo sapiens were generated based on experimental evidence of interactions and co-expression, with a minimum interaction score set >0.400 (medium). The assembled networks were then imported into the Cytoscape v.3.10.0 software to calculate network statistics [52]. Two critical parameters were considered: (1) betweenness centrality, which reflects the frequency with which a node lies on the communication paths between other nodes, indicating that nodes with high betweenness centrality may be crucial for regulating information flow; and (2) closeness centrality, which measures how efficiently information spreads from a central node to others [53]. A comparison between the GO and HPO networks was performed using the DyNet application of Cytoscape v.3.10.0.

### 4.3. Selection and Analysis of Gene Expression Data

Gene expression datasets were obtained from the Gene Expression Omnibus (GEO) repository [54] using the following search strategy: ((cardiac OR heart OR cardio) AND (anomaly OR defect OR malformation)), filtered for Homo sapiens. The data from each study were collected using the GEO Scraper script and selected by two authors. The inclusion criteria were as follows: studies of gene expression in human cells, tissues, or samples from patients diagnosed with CHD, conducted using microarray or RNA-seq technologies. Studies without raw data available in the GEO, knockout studies, studies with only four samples (n = 4) in the case and/or control groups, and studies without a CHD diagnosis were excluded from the analysis.

For microarray studies, the datasets were downloaded manually and analyzed in the R language using Robust Multi-array Average (RMA) with the affy package [55]. Differentially expressed genes were calculated using the limma package [56]. RNA-seq data followed the pipelines described by Conesa et al., 2016 [57]. Sequence read archives (SRAs) were uploaded into the Galaxy platform [58] using the fastq-dump tool [59]. The quality of the sequences was assessed using FastQC (https://www.bioinformatics.babraham.ac.uk/projects/fastqc/; accessed on 7 September 2024) [60], followed by sequence alignment to the reference genome GRCh38 (hg38 canonical) using Bowtie2 [61], and transcript counting with featureCounts [62]. In R, normalization was performed using Trimmed Mean Normalization (TMM), and differentially expressed genes were analyzed using the edgeR package [63]. Genes with a logFC > |1| and an adjusted *p*-value < 0.05 were considered significantly differentially expressed.

Using the STRING tool, we analyzed the combined GO and HPO networks together with the differentially expressed genes, as previously defined. Each category was individually integrated with the GO/HPO network, and the resulting networks were applied to Cytoscape v.3.10.0. We evaluated the interaction between the genes found in GO/HPO and the differentially expressed genes, and identified if any gene presented gene ontology and/or phenotypic characteristics associated with CHD.

### 4.4. Enrichment Analyses

The differentially expressed genes identified through DGE that were not common with the gene lists obtained from the GO and HPO were evaluated concerning ontologies and signaling pathways. The GO repositories and the Kyoto Encyclopedia of Genes and Genomes (KEGG) pathways were accessed using the clusterProfiler package [64], using over-representation analysis. For this analysis, the studies were divided into four groups based on CHDs: (1) isolated CHD studies (Tetralogy of Fallot); (2) one isolated CHD study (Hypoplastic Left Heart Syndrome); and (3) CHD studies associated with Down syndrome. The genes identified in the HPO that were not common with those found in the GO were also subjected to the same analysis. Biological processes, molecular functions, and cellular components were the types of ontologies accessed in this analysis. A summary of all databases used in this study can be seen in Table 3.

## 5. Conclusions

The approach used in this study allowed us to integrate several genes previously described for congenital heart disease (CHD), motivating us to explore new interactions based on the phenotypes analyzed from public datasets. The availability of these data allowed us to identify and characterize 2513 genes in six studies analyzed while seeking to understand the biological processes involved and the interactions between these genes. We identified genes that still need to be directly described in the literature on CHD. Knowing that the heart is the first functional organ to develop, in parallel with other biological processes that occur during embryogenesis, it is necessary to understand cellular differentiation in cardiogenesis, especially when we seek to understand the varied phenotypes of CHD and the repercussions of the disease. Therefore, the data presented will contribute to a better understanding of CHD and provide valuable insights for future research.

## Figures and Tables

**Figure 1 ijms-25-12052-f001:**
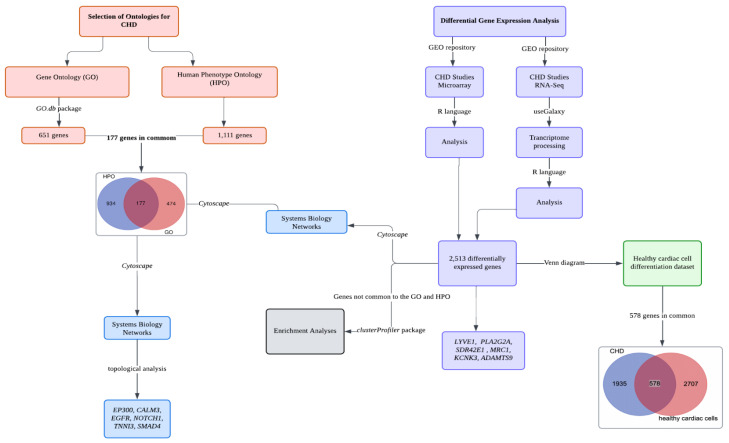
Schematic research strategy comprising systems biology analysis, differential gene expression analysis, and enrichment analyses.

**Figure 2 ijms-25-12052-f002:**
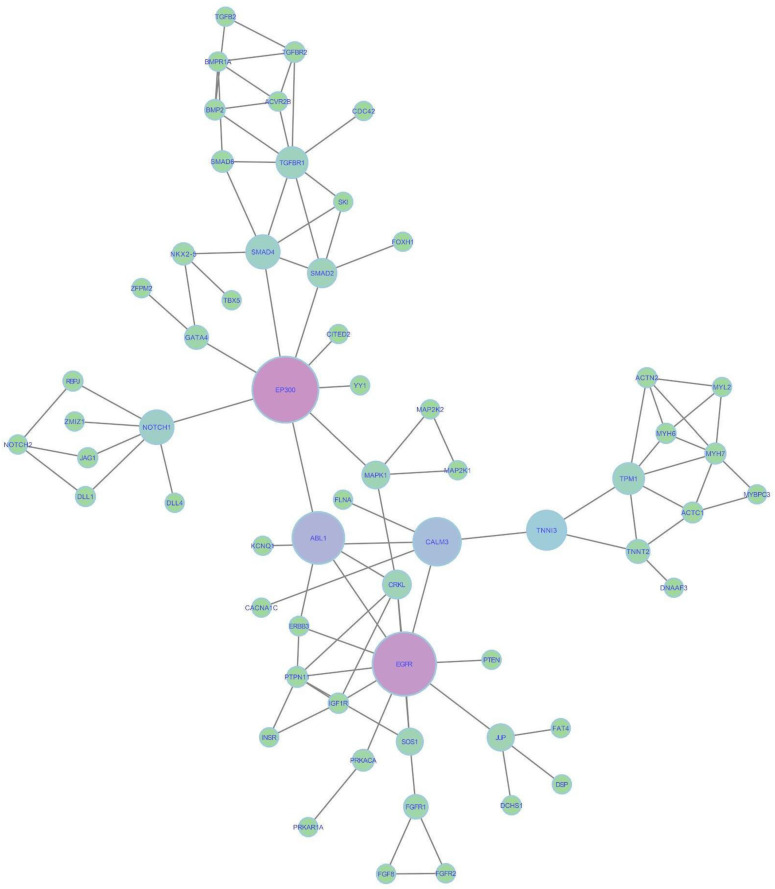
A network comprising experimental evidence of protein–protein interactions, considering genes in common between the GO and HPO repositories. The size of each node reflects the speed of information flow, with larger nodes indicating a faster flow. Pink and purple nodes represent genes with the highest levels of betweenness and closeness centrality, acting as critical control points for the flow and the speed at which information is relayed to other genes in the network. The green nodes—NOTCH1, TNNI3, and SMAD4—exhibit significant closeness centrality, playing key roles in mediating information flow within smaller groups of genes in the network.

**Figure 3 ijms-25-12052-f003:**
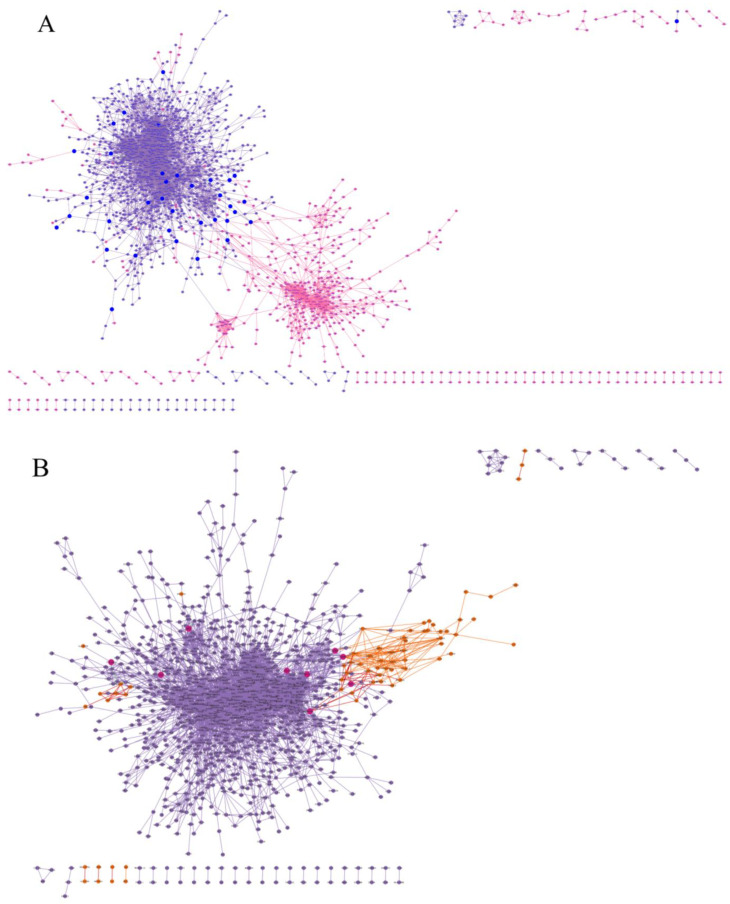
A network of interactions between differentially expressed genes from each study and genes associated with the Gene Ontology (GO) and Human Phenotype Ontology (HPO). (**A**) Purple nodes represent GO/HPO genes, pink nodes represent genes from Tetralogy of Fallot studies, and blue nodes represent common genes identified and already described for gene ontologies and phenotypes associated with CHD. (**B**) Purple nodes represent GO/HPO genes, orange nodes represent genes from the Hypoplastic Left Heart Syndrome study, and magenta nodes represent common genes identified and already described for gene ontologies and phenotypes associated with CHDs. (**C**) Purple nodes represent GO/HPO genes, and green nodes represent genes associated with Trisomy 21/CHD. In this case, there was no interaction between GO/HPO genes and differentially expressed genes, and no genes in common between the databases.

**Table 1 ijms-25-12052-t001:** The proportion of differentially expressed genes considering downregulated genes, upregulated genes, and shared genes in the GO and HPO repositories.

Study	Study Type	CHD	Controls (N)	Cases (N)	DGE (N) ^1^	Upregulated	Downregulated	GO (N) ^2^	HPO (N) ^3^	GO + HPO
GSE196443	RNA-Seq	Trisomy 21/CHD	5	5	23	23	0	2	0	0
GSE217557	RNA-Seq	Trisomy 21/CHD	32	50	13	12	1	0	0	0
GSE36761	RNA-Seq	ToF ^4^	7	22	2228	727	1501	65	59	13
GSE132401	RNA-Seq	ToF ^4^	5	5	111	69	42	6	5	4
GSE141955	Microarray	ToF ^4^	6	9	35	2	33	0	0	0
GSE23959	Microarray	HLHS ^5^	6	10	184	130	54	9	9	2

^1^ Differentially expressed genes identified. ^2^ Gene Ontology. ^3^ Human Phenotype Ontology. ^4^ Tetralogy of Fallot. ^5^ Hypoplastic Left Heart Syndrome.

**Table 2 ijms-25-12052-t002:** Genes that may contribute to the development of CHD and its functions.

Gene	Analysis Source	Gene Function
*EP300*	Systems Biology Networks	Chromatin binding and transcription coactivator activity.
*CALM3*	Systems Biology Network	Calcium ion binding and protein domain specific binding.
*EGFR*	Systems Biology Network	Identical protein binding and protein kinase activity.
*NOTCH1*	Systems Biology Network	DNA-binding transcription factor activity and sequence-specific DNA binding.
*TNNI3*	Systems Biology Network	Protein kinase binding and protein domain specific binding.
*SMAD4*	Systems Biology Network	DNA-binding transcription factor activity and sequence-specific DNA binding.
*LYVE1*	DGE	Signaling receptor activity and hyaluronic acid binding.
*PLA2G2A*	DGE	Calcium ion binding and phospholipase A2 activity.
*SDR42E1*	DGE	Oxidoreductase activity, acting on the CH-OH group of donors, NAD or NADP as acceptor and 3-beta-hydroxy-delta5-steroid dehydrogenase activity.
*MRC1*	DGE	Signaling receptor activity and mannose binding.
*KCNK3*	DGE	Protein homodimerization activity and obsolete protein C-terminus binding.
*ADAMTS9*	DGE	Metalloendopeptidase activity and endopeptidase activity.

**Table 3 ijms-25-12052-t003:** Description of databases used in this study.

Database	Description	Main Features	Purpose of the Study	Reference
Gene Ontology (GO)	It provides structured information about genetic functions, serving as the basis for computational analysis of large-scale molecular biology and genetic experiments.	Data availability in three categories: biological process, molecular function, and cellular component.	Identify the available ontologies for the development of CHD.	The Gene Ontology Consortium, 2023 [65]
Human Phenotype Ontology (HPO)	It provides an ontology of clinically relevant phenotypes, disease phenotype annotations, and the algorithms that operate on them. The HPO can be used to support differential diagnoses, translational research, and a range of applications in computational biology, providing the means to compute clinical phenotypes.	Describes phenotypic abnormalities in human diseases.	Identify phenotypes associated with CHD.	Gargano et al., 2024 [66]
Gene Expression Omnibus (GEO)	A public repository for high-throughput gene expression data, where you can access datasets from multiple organisms and biological conditions.	It includes publicly accessible gene expression, microarray, and RNA-Seq data.	Investigate gene expression profiles related to CHD.	Barrett et al., 2013 [67]
STRING	It systematically integrates protein–protein interactions from diverse sources, including the scientific literature, experimental databases, and computational predictions.	Data are curated from diverse sources: scientific literature, computational interaction predictions, coexpression, conserved genomic context, databases of interaction experiments, and known complexes/pathways from curated sources.	Identify relevant interaction networks for CHD-related genes using experimental data and coexpression.	Szklarczyk et al., 2023 [68]
Kyoto Encyclopedia of Genes and Genomes (KEGG)	A database for representing and analyzing biological systems, with maps of metabolic and signaling pathways, cellular interactions, and disease pathways.	Includes information on genes and proteins, disease pathways, drug information, and integration with other databases.	Identify biological pathways involved in CHD and their associated genes.	Kanehisa et al., 2024 [69]

## Data Availability

The study only used data publicly available in databases and repositories. All the results generated are fully available in the Appendix A.

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
