# Peer review of "Bioinformatic Multi-Strategy Profiling of Congenital Heart Defects for Molecular Mechanism Recognition"

_ijms, 2024, doi:10.3390/ijms252212052_

Round 1
Reviewer 1 Report
Comments and Suggestions for Authors
The article by Oliveira et al is a bioinformatics analysis of differentially expressed genes involved in Congenital heart disease from publicly available information and databases. As such the study is not providing any novel results except for re-analysis of gene sets available and arrive at certain set of genes involved in immune response and energy production as some relevant ontologies of CHD. The analysis seems fine, and the results presented are clear. However, without experimental evidence proving these genes in the cause of CHD the significance of such an analysis is not clear. This is a limitation of the study as mentioned in the discussion.
Reviewer 2 Report
Comments and Suggestions for Authors
Thanks to the authors for their courage in trying to present to IJMS readers “Bioinformatics multi-strategy profiling of Congenital Heart 2 Defects (CHD) for molecular mechanisms recognition”
I learned a lot from reading the article, but it's very difficult to understand, without constantly going back and forth with outside references other than the article itself.
I therefore recommend that you present in a few words each time you use a new database what its main characteristics are. This could also be done in a table of the article which would interest the readers and promote the circulation of the article.
Some of your statements are very surprising and need to be explored before being written. For example
Line 100 – “Of the genes identified in HPO, 934 (84%) have not previously been associated with coronary heart disease, however, they are involved in other biological processes such as peroxisome maintenance and organization, the assembly and organization of cilia and microtubules. transport and travel »
I suggest a hypothesis, many CHD are fatal during embryonic and fetal life, those we see after birth are the survivals, the "tip of the iceberg" which may be a reason for the discrepancy you have observed. Is it possible for you to introduce the period of life into your algorithm?
Another example of the same problem stands Line 264 “Using systems biology, we integrated the genes identified in GO and HPO to provide new insights into CHD, revealing genes that might play key roles in the flow of biological information. The EP300, critical for cell regulation and differentiation through chromatin remodeling, already has pathogenic variants associated with ToF [28]. EGFR encodes tyrosine kinase receptors and is essential for cardiac cells’ development, specialization, and differentiation, particularly in regulating human aortic valve embryogenesis [27, 28]. CALM3 encodes a highly conserved protein expressed in the heart, regulating various ion channels in cardiac cells and being involved in several biological processes, such as muscle contraction, inflammation, metabolism, and immune responses [29]. EP300, EGFR, and CALM3, along with other identified genes, have already been linked to CHD and are crucial to the disease's development; however, further exploration of the genetic regulation of cardiogenesis is still needed.”
In the reference 28 you cite, the authors insist on the embryonic and fetal lethality of ErbB receptors mutations“.
Line 112 “Based on this analysis, EP300, CALM3, and EGFR had the highest betweenness and closeness centrality levels, meaning they have high and fast information flow. Additionally, the genes NOTCH1, TNNI3, and SMAD4 emerged as significant mediators of the information flow to other genes within the network”.
Please better explain the criteria that distinguish between high and rapid information flow from mediators. Furthermore: In Figure 2, Notch1 and TNNI3 are clearly identified as mediators of information, this is less obvious for SMAD4. This could be explained in the text or in the caption.
Line 244 “Hence, using gene ontology overrepresentation analysis, we found that most of these genes are Involved in immunological processes, energy generation, secondary metabolite production, and cellular communication, which are fundamental to heart formation. Nevertheless, the absence of experimental validation for the genes presented constitutes a limitation of the current study”.
Isn't this expression too general? This only means that a combination of hypomorphic alleles of “housekeeping genes” constitutes the first fragile foundation of an unstable edifice. This is a general problem related to the results of the bioinformatics strategy you are using. Quickly, you recruit thousands of genes.
Figure 3 - I don't understand the metrics of this interaction network. This needs to be better explained in the text and in the caption.
At last L 60 the sentence is wrong! “Genetic contributions to the development of CHD vary widely, i.e., aneuploidies are diagnosed in 35 to 50% of newborns with Trisomy 21, in 60 to 80% with Trisomy 18 and 13, and in 33% with Monosomy X [12, 13].”
I'll let you figure out what's wrong with the sentence, but it comes early in the article and it gives the wrong impression which is a shame, because there's a lot of good stuff in the article.
Comments on the Quality of English LanguageAt last L 60 the sentence is wrong! “Genetic contributions to the development of CHD vary widely, i.e., aneuploidies are diagnosed in 35 to 50% of newborns with Trisomy 21, in 60 to 80% with Trisomy 18 and 13, and in 33% with Monosomy X [12, 13].”
I'll let you figure out what's wrong with the sentence, but it comes early in the article and it gives the wrong impression which is a shame, because there's a lot of good stuff in the article.
Round 2
Reviewer 1 Report
Comments and Suggestions for Authors
The article by Oliveira et al is a bioinformatics analysis of differentially expressed genes involved in Congenital heart disease (CHD) from publicly available information and databases. The study provides interesting results with analysis of gene sets available and arrive at certain set of genes involved in immune response and energy production as some relevant ontologies of CHD. The analysis seems fine, and the results presented are clear.
The revisions are satisfactory. In the near future, it would be interesting if the finding reported in this bioinformatic analysis can be validated by experimental approach.
Reviewer 2 Report
Comments and Suggestions for Authors
Thank you to the authors for their efforts to clarify their practices and thoughts in order to make them understandable to a broader community than the strict bioinformatics community. I was pessimistic about your possibilities of doing this, and it was a pleasant surprise for me to read your responses to all my comments.
There are some points that remain difficult to understand, but these are now more points of debate than errors or unnecessary jargon.
Furthermore, your conclusions remain cautious and I consider your article to be a useful contribution to the scientific and medical community. So congratulations.